# Coping with a Disruptive Life Caused by Obstetric Fistula: Perspectives from Malawian Women

**DOI:** 10.3390/ijerph16173092

**Published:** 2019-08-26

**Authors:** Josephine Changole, Viva Thorsen, Jone Trovik, Ursula Kafulafula, Johanne Sundby

**Affiliations:** 1Department of Community Medicine and Global Health, University of Oslo, P.O. Box 1130 Blindern, N-0318 Oslo, Norway; 2Department of Clinical Science, University of Bergen, P.O. Box 7804, 5021 Bergen, Norway; 3Department of Obstetrics and Gynecology, Haukeland University Hospital, National Treatment Center for Gynecological Fistula, P.O Box 1400, 5021 Bergen, Norway; 4Department of Maternal and Child Health, University of Malawi, Kamuzu College of Nursing, P.O. Box 415 Blantyre, Malawi

**Keywords:** fistula, incontinence, Malawi, Bwaila, coping, concealing, problem-focused, emotional

## Abstract

*Background*: The main symptom of obstetric fistula is urinary and or fecal incontinence. Incontinence, regardless of the type is debilitating, socially isolating, and psychologically depressing. The objective of this study was to explore the strategies that women with obstetric fistula in Malawi use to manage it and its complications. *Methods*: A subset of data from a study on experiences of living with obstetric fistula in Malawi was used to thematically analyze the strategies used by women to cope with their fistula and its complications. The data were collected using semi-structured interviews. Nvivo 10 was used to manage data. *Results*: Participants used two forms of coping strategies: (1) problem-based coping strategies: restricting fluid intake, avoiding sexual intercourse, using homemade pads, sand, corn flour, a cloth wreathe and herbs, and (2) emotional-based coping strategies: support from their families, children, and through their faith in God. *Conclusion*: Women living with incontinence due to obstetric fistula employ different strategies of coping, some of which conflict with the advice of good bladder management. Therefore, these women need more information on how best they can self-manage their condition to ensure physical and emotional comfort.

## 1. Introduction

Urinary incontinence, the involuntary loss of urine, can affect people of all ages, social, and cultural backgrounds, however the causes may differ [1]. Obstetric fistula is a hole between the birth canal and bladder or rectum caused by prolonged, obstructed labor, which has not been treated in time [2]. This hole causes the woman to leak urine and/or feces uncontrollably through her vagina. Obstetric fistulas may also develop due to complications from caesarean sections or perineal tears [3,4]. According to the World Health Organization (WHO), more than two million girls and women in Sub-Saharan Africa and South Asia are believed to be suffering from obstetric fistula [5]. However, more recent pooled estimates from a comprehensive systematic review suggest that just over one million young girls and women may have a fistula in sub-Saharan Africa and South Asia, and that there are more than 6000 new cases per year within these two regions [6]. Given the devastating consequences of fistula for women and their families, this represents a very substantial burden. The actual prevalence of obstetric fistula in Malawi is currently 1.6 per 1000 women of reproductive age (15–49) [7]. Anecdotal reports show that 20,000 (approximately 5 per 1000) women are currently living with a fistula [8]. The Malawi Demographic Health Survey (MDHS) 2016 [9], indicated that one per cent of 24,562 women (*n* = 246) who participated in the survey had experienced obstetric fistula symptoms between 2010 and 2015. Nevertheless, all these figures are based on self-reported incontinence, which could also result from other conditions, consequently overestimating the prevalence; women may also feel embarrassed by their condition and not report their incontinence leading to underestimation of the prevalence. In other words, the actual prevalence is unknown.

Obstetric fistula can affect a woman in many ways. A systematic review of qualitative studies on consequences of obstetric fistula [10] reported that obstetric fistula brings forth physical, socio-economical, and psychological problems in the life of the affected woman. For example, due to being in constant contact with urine and/or feces, the woman may develop genital sores, which makes her movements difficult due to pain [10,11]. The wetness and the unpleasant smell of urine bring shame, ridicule, self and social isolation, stigmatization, and depression [10,11,12]. The urine and/or fecal leakage may interfere with sexual relationships leading to abandonment and, in extreme cases, divorce [10,11]. Furthermore, because of self and social isolation, the woman may not participate in income-generating activities, leading to poverty [10,11].

Obstetric fistula may be treated with an 80–98 percent closure success rate [13,14,15], implying that women may regain total or partial control of their urinary function, thereby potentially regaining their dignity in the process [8]. Unfortunately, due to shortage of resources and skilled surgeons to perform such repairs [16], women in low-income countries, such as Malawi, continue to suffer for many years, or indeed, for the rest of their lives. Additionally, even when treatment is available, women with obstetric fistula may conceal their condition from others, including health workers [8,17], thereby further prolonging their suffering.

Women with fistula do not passively suffer. They use coping strategies to deal with their condition [18]. In problem-focused coping, women use available resources to manage or change the problem-causing stressor. This entails appraising and reappraising the situation, identifying resources, and planning activities to deal directly with the condition. In emotional-focused coping it is not about changing or managing the stressor itself, but instead figuring out how to become less emotionally affected by the stressor so that the impact of the condition is experienced differently [18].

In Malawi, previous studies on obstetric fistula focused primarily on prevalence, surgical repair, experiences, and outcomes [7,8,13,19,20,21], with scant attention to coping with obstetric fistula. Hence, this study aimed to explore how Malawian women with obstetric fistula coped with fistula incontinence and its associated stress, hardship, and social consequences. The insights gleaned from exploring these strategies may inform the development of new programs (or improve existing ones) to improve their wellbeing, and restore hope, their identity, and dignity.

## 2. Methods

This paper is based on a subset of data of a qualitative study [22]. The details of the setting and data collection methods used for this study were previously published [8]. In sum, the study was both institutional and community based. In the central region of Malawi, 20 women diagnosed with obstetric fistula were purposively recruited from the Bwaila Fistula Care Center (BFCC). BFCC was selected because it is the only fistula care center in the country, and the researchers were ensured access to women affected by fistulas. Five women were recruited through snow ball sampling [23] in order to reach out to women with fistula in the communities who had not yet sought treatment at BFCC due to lack of knowledge, fear of stigma or other reasons. The first author (JC) conducted all of the face-to-face interviews using a semi-structured interview guide. Face-to-face interviews were considered the most appropriate data collection tool because they enable flexible, in-depth exploration of the coping experiences and allow participants to freely express themselves, without being limited by predetermined responses, as is the case with questionnaires [22]. The interviews were conducted in a doctor’s private office, which was set apart for this purpose to ensure privacy, and were conducted at the participant’s convenience, in order not to interfere with the doctor’s plans for her care. Interviews with women in the communities were also conducted at their convenience, at a time and place of their choice to ensure privacy. Data were collected until saturation was reached, that is, when no new insights could be drawn from subsequent interviews.

### 2.1. Data Analysis

All audio interviews were transcribed verbatim and translated from Chichewa to English by the first author (JC) and three research assistants. The English transcripts were used for analysis. A thematic approach [24] was used to analyze the data. A detailed description of the data analysis was previously published [8]. Data on coping strategies were read and reread to become familiarized with the data. The data were then coded and categorized into major themes and subthemes using Nvivo 10, a qualitative data analysis computer software package, developed by QSR International Pty Ltd., Australia. Several rounds of discussion between three of the co-authors (JC, VT, UK) took place to strengthen the credibility and integrity of the findings [25]. To ensure confirmability, three of the co-authors (JC, VT, UK) reflected on the data, discussed differences in the interpretation of data, and agreed on the categorizations [26].

### 2.2. Ethical Consideration

Ethical approval was obtained from the College of Medicine Research Ethics Committee (COMREC) on 28 May 2015; reference number P.03/15/1711 and was registered with the Norwegian Social Science Data Services (NSD) in Norway, reference number 43620 dated 25 June 2015. Voluntary participation was emphasized. Participants gave both oral and written consent. Participants were given an equivalent of $2 for transport either to the hospital for those recruited through snowball sampling or back to their homes if recruited from the care center. This was done at the end of the interview to avoid undue incentivization [27]. Pseudonyms are used in this report to protect participants’ identities and to ensure confidentiality.

## 3. Results

### 3.1. Participants’ Characteristics

A total of 28 women clinically diagnosed with obstetric fistula were invited to participate in the study. Three women declined. Of the 25 women who participated, six did not know their age. Of those who did, their mean age was 36 (range = 16–67 years). The majority of participants (14) had no formal education. The majority (15) were multipara and had uncomplicated vaginal birth(s) prior to the birth that caused their fistula. Thirteen (13) were still married, while five (5) were divorced, two (2) were single, and the rest were widows (5). All but two participants delivered babies that were stillborn. The median years of living with fistula was 13 (range = 1 month–47 years). Detailed information on the characteristics of participating women with fistula has been previously published [8].

### 3.2. Advent of A ‘New Normal’

Testimonies from participants illustrated how their lives underwent an instant and dramatic change due to the consequences of fistula. Their previous normal lives were devastatingly transformed to what some women referred to as ‘slavery’. In order to maintain a sense of normalcy, the participants used both problem-focused and emotional-focused coping. Their narratives illustrated several pathways that they took to manage their incontinence, with the majority following a similar pathway of first becoming aware of the condition, then continuously searching for a cure, next accepting the disrupted life of living with incontinence, eliciting spouse and family support, and over time strengthening their faith in God. Figure 1 illustrates the pathway constructed, based on the experiences of managing and coping with their condition that they shared.

#### 3.2.1. Managing Urine/Fecal Leakage Throughout All Facets of Life

Having failed to obtain help from different sources [8], most participants accepted their situation and started to adjust their lifestyles to accommodate the disruptions caused by incontinence. Participants used different strategies to contain and restrict the leakage of urine or feces, to prevent offensive smell, and to treat sores in order to maintain social interactions, and to achieve physical comfort as much as possible. These are considered problem-focused coping strategies. The strategies are described below.

#### 3.2.2. Containing and Concealing Urine and/or Fecal Leakage

Containing and concealing were usually the initial reactions to the leakage. Participants mainly used homemade pads folded from worn out clothing or blankets. Some participants reported using these materials together with water proof materials such as plastic sheets, to ensure urine and or feces never permeated through their clothing. See Figure 2, for example.
“*I found a plastic paper, made holes on it, one on each side. Then, I took the paper and put some strings on each side, so I would put on a cloth pad and then put on that paper under it. So if the paper does not leak, I would stay well. And some people did not even know that I get wet.*”(Nabiyeni, 57 years, para 2, 21 years with urinary incontinence)

Same strategies were used by participants with fecal incontinence, as one summarized:
“*Urine just comes on its own, and sometimes when I want to defecate, feces come out through my vagina, and sometimes through my anus. But I see that it has recently changed, it is not as heavy as before. So I just put on a cloth pad on my underwear, wear a half petticoat, a skirt, and on top of that a Chitenje (wrapper). So I could walk freely without people noticing that I get wet.*”(Nanyoni, 20 years old, para 1, 4 months living with urinary and fecal incontinence)

##### While Traveling

Participants expressed how the condition had limited their movements. They only traveled when it was extremely necessary to do so. For example, when they had to attend a funeral of a close relative. In such cases, participants described the preparations they took to ensure they passed as normal wherever they went. These preparations involved carrying extra clothing pads, perfumed soap, and wrappers (Zitenje) that enabled them to change into these when they became wet with urine.
“*When I am travelling long distances, I carry spare cloth pads in my handbag.*”(Namaisa, 36 years old, para 2, 15 years living with fistula)

##### While Having Sex

Participants who were still sexually active despite the leakage of urine/feces described how they controlled the leakage during sexual intercourse:
“*So where I sleep I spread a plastic sheet first and then a mat, and on top of the mat a cloth to lie on. … we can have sex without any problem, but when he withdraws, that is the moment I make sure that the pieces of cloths are under my buttocks to collect the urine, so that urine does not soak through to the mat.*”(Nasiketi 33, para 3, 12 years with urinary incontinence)
“*The time we used to have sex it was a problem, I had to dry myself with a cloth every now and then for it to happen. The cloth I used to dry myself would be so wet, but we would do it anyway, really.*”(Nachanza, 41 years old, para 8, 19 years with urinary incontinence)

One participant opted for non-penetrative sex.
“*… he could just do it outside, did not enter me, but he did not complain about anything. I guess the only difference was when he was with his other wife that he would compare the satisfaction, but he never expressed dissatisfaction.*”(Nabiyeni, 57 years old, para 2, 21 years with urinary incontinence)

Some participants believed sexual activity made the leakage of urine excessive, and because they felt embarrassed by those experiences, they limited or avoided sex altogether.
“*When I developed this problem, and was divorced, my plan was to stay alone for a while without a man. Because I was thinking that, maybe the problem is worsening because of being with a man (having sex), really.*”(Nasikelo, 32 years old, para 1, 12 years with urinary incontinence)

##### While Sleeping

Two participants described extraordinary methods that they used to prevent urine from spreading out as they slept.
“*My mother would spread out sand on the floor, and then she would put the mat on top of the sand. Then I would sleep on top of that mat. She did this to control the flow of the urine, to prevent it from spreading out to a great area. She would even run out of ideas and reach the extent of digging a small hole in my room, and I would put my buttocks on that hole, so that the water drains in the hole.*”(Nasawa, 52 years old, 27 years with urinary incontinence)
“*When the urine was so profuse, my aunt would make a cloth wreathe for me to place my bottom on. She would wreathe the cloth and have me sit on it and I’d sit on it and sleep. When that one is soaked, she would take it the following day and go wash it, and make another one for me.*”(Nabiyeni, 57 years, para 2, 21 years with urinary incontinence). See Figure 3 below, for example.

##### While Farming

Most of the participants were subsistent farmers, and the urine leakage did not stop them from doing any of their farm work. One participant described how she just ignored the leakage as she went about working on her fields.
“*When I go to the field to clear and till my field, I did not bother about it (leakage). I could do my hoeing; “let it leak, I don’t care, as long as I do my work.*”(Nadzimbiri 37, para 5, 12 years with urinary incontinence)

### 3.3. Restricting the Flow

Participants used different ways to control urine and or fecal leakage. A couple of participants restricted their intake of food, water and other fluids in preparation for travel to other places to reduce or avoid accidental leakages that could involuntarily disclose their condition.
“*If I want to go to church the following morning, I avoid all the eating and drinking even when l feel hungry; I control myself. So when I do not eat or drink, the leaking is minimal, it just drips little by little. But the problem is that the urine is so hot and painful, but all the same, I manage to travel a long distance and come back without any problem.*”(Nadzimbiri, 37 years old, para 5, 12 years with urinary incontinence)

Some participants adopted a sitting position that allowed them to trap the urine and give them time to rush to the toilet to release it before it dripped down their legs.
“*(laughing)... I would sit like this,* [crisscrossing her legs] *then after sometime I would get up and go somewhere, where I would just spread my legs and let go.*”(Namoyo, 16 years old, para 1, 1 year with urinary incontinence)

### 3.4. Managing the Odor

Odor of urine and/or feces was one of the most bothersome effects of incontinence expressed by our participants. The most common strategies used by participants was frequent bathing, washing, and changing of cloth pads, as well as avoiding places with no accessible water and pit latrines for changing [8]. Other strategies included using perfumed soaps and talcum powder, and sitting at strategic points when in public, as shared by one participant:
“*I always sit following the direction of wind, so that the smell blows away from the rest of the people.*”(Nalike, age unknown, para 6, 4 years with urinary incontinence)

### 3.5. Managing Sores Caused by Urine Leakage

Participants reported developing sores from being in constant contact with urine, and friction from rubbing against wet pads and wet thighs. Methods used to prevent and treat the sores included washing the sores with cold water, applying talcum powder, body lotion, and staying indoors, naked, to expose the sores to air for speedy healing.
“*… I would apply body lotion on the sores, lock the door, and lie on my back, with my legs flexed and spread like this [lying on her back and flexing and spreading her legs in demonstration] I would spend the whole day, sometimes two or three days, indoors, naked. If people asked about me, my sister would just tell them that ‘Don’t worry she is around, but is busy’.*”(Nadzimbiri, 37 years, para 5, 12 years with urinary incontinence)
“*…There are times when I pass out hot water [urine] and sometimes cold water [urine]. When I pass out hot water, I don’t feel well, and that means I have to bath frequently in cold water to ease the discomfort.*”(Nasibeko, 32 years, para 6, 16 years with urinary incontinence)

Water scarcity made it extremely difficult for some participants to manage their sores. One participant bemoaned her situation as follows:
“*When they [sores] start itching, I would take water and pour on them, but when water is not available, then I would just continue scratching, but in the end it will be fire of sores. Ii! It was very pathetic, my friend, it was such suffering, especially the itching, ii! You had to scratch and scratch, until blood came out, and that meant sores. Now when I came here [Bwaila hospital], they give us skin protection cream. I wish they gave me that protection cream back at our health center, I wouldn’t have had those sores.*”(Nabiyeni, 57 years, para 2, 21 years with urinary incontinence)

Another participant used antibiotic capsule powder and traditional herbs to treat her sores.
“*To manage them (sores) I would buy the yellow and red capsule and put the powder contents on the sores; so painful! I would also use some herbs found in our area they call it chamasala, also very painful. You get better two, three days, and they reappear again, iii! No peace at all, no peace. It is slavery.*”(Nasawa, 52 years, para 2, 27 years with urinary incontinence)

While another participant used corn flour:
“*When I develop sores, I would apply white maize flour on the sores. And I will just be staying in the house, not going out. I would not put on any pads or underwear. I will just let the urine flow freely. Because cloth pads also cut on your skin and worsen the sores…*”(Nasiketi, 33 years old, para 3, 12 years with incontinence)

### 3.6. Emotional-Focused Coping

The majority of participants likened their lives to that of a slave and expressed that they lacked peace all of the time. However, they all demonstrated a degree of resilience with their condition. When asked what kept them going, participants described several factors, including being able to keep their condition secret, which was also a problem focused strategy. Other strategies included receiving spouse and family support, being motivated by having children to nurture, and strengthening their faith in God.

#### 3.6.1. Spouse and Family Support as Source of Encouragement

Most participants reported feeling encouraged by the support they received from their spouses, family members, and friends. Participants reported feeling especially encouraged when they were accepted despite their condition. They also reported receiving material and financial support particularly from their spouses and family members.
“*My relatives loved me. For example, among my relatives, there was no one who spoke ill of my condition. People from my husband’s side would say, “This is just a disease. After all, when she was coming here, we did not know that she would end up like this.*”(Namaisa, 36 years, para 2, 15 years with urinary incontinence)

#### 3.6.2. Child Rearing as a Source of Motivation

Some participants were encouraged by the fact that they had children who needed their support and care.
“*…When I think about my condition, and again, when I look at my children…I say, “What of my children? And that encourages me.*”(Nasiketi, 33 years, para 3, 12 years with urinary incontinence)

#### 3.6.3. Having Faith in God as a source of Encouragement

The majority of our study participants were Christians, so reported being strengthened by their faith in God.
“*At first, I was broken hearted, because it was my first pregnancy. I was worried and wondering what my future would be like. But God has a way of consoling people’s hearts and giving them strength, and so he strengthened me. As time went by I accepted my condition, that ‘What has happened, has happened, let it be.’ So, I was just staying like that.*”(Namaisa, 36 years old, para 2, 15 years living with fistula)

A few participants disclosed their condition to members of their women’s religious groups, so they got spiritual support in terms of prayers for their healing and encouragement to seek medical care.
“*I told my friends at our catholic women organization’s gatherings about my condition, and they told me that they would be praying for me, and they were indeed praying for me, and that encouraged me.*”(Nabanda, 55 years, para 4, 21 years with urinary incontinence)

## 4. Discussion

Findings indicate that despite the disruptive nature of obstetric fistula, women cope and normalize their lives as much as possible. Women used both problem-focused and emotional focused coping strategies [18]. Findings also indicate that poverty and lack of information about self-care of their symptoms dictated the way they managed some of the physical consequences of fistula. Apart from the use of sand and a cloth wreathe, the strategies reported in this study are consistent with findings from previous studies [28,29,30,31]. In the following sections we discuss these findings in relation to previous studies.

### 4.1. Problem-Focused Coping

#### 4.1.1. Containing and Concealing Urine and/or Fecal Leakage

Strategies used by our study participants to contain urine and or fecal leakage are similar to those synthesized by St. John and colleagues [31]. However, the quality of the absorbent materials used was extremely different from those reported in the above study. That is, our study participants mostly depended on pads made of old fabric materials, such as pieces of old clothing or pieces of worn out blankets, while St John et al.’s study noted adult diapers and incontinence pants, which are currently not readily available in Malawi. Even if such products were readily available in the country, it is doubtful that our study participants could afford them. Nevertheless, use of old clothing pads has previously been reported in similar settings [28,32]. Encouraging though, is the reusable sanitary pads project currently being implemented at BFCC by the Freedom from Fistula Foundation (FFF) [33]. These products not only help women with incontinence, but also all menstruating girls and women who cannot afford disposable sanitary pads.

To the authors’ knowledge, the use of sand, digging a hole in the bedroom, and sitting and lying on a cloth wreathe have not been reported before. Because women with fistula are predominately from poor settings and could benefit from such methods as they await surgical repair, further investigations are warranted.

#### 4.1.2. Concealing Leakage

In Malawi, engaging in social activities such as weddings and funeral ceremonies is very important for social inclusion and belonging. It is not surprising that concealment was used as a coping strategy as previously reported [28,30]. With concealment tactics, one would be able to maintain social continence, self-identity, self-respect, and avoid stigmatization [8]. However, self- isolation, avoiding social gatherings and public places, and nondisclosure may negatively affect health care seeking [17,34], as one may miss information about available care and social support. Furthermore, self-imposed restrictions and limited social engagements, may lead to loss of friendships, loss of economic opportunities, increased loneliness, and boredom [35], which may further result in depression [12].

#### 4.1.3. Restricting Urine and or Fecal Leakage

The East, Central, and Southern African Health Community (ECSA-HC) and Fistula Care/Engender Health, and International Federation of Gynecology and Obstetrics (FIGO) and its partners recommend up to 5 L of fluids per day for women with fistula to avoid urinary and skin-related problems [36,37]. However, some women in our study reduced food and fluid intake to reduce leakage. Such restrictions may negatively impact kidneys and bladder functions [38]. Also, restricting fluid intake may lead to chronic dehydration, and possibly kidney and vaginal stones [37]. Additionally, concentrated urine may cause skin irritations on the genitalia, inner thighs, and buttocks, leading to dermatitis and excoriation on those areas [38]. Irritated skin is conducive to bacterial growth, which may cause urinary tract and skin infections [38]. Health care providers should use every opportunity to educate, counsel, and encourage affected women to drink recommended amounts of fluids, preferably water, to minimize the above sequelae.

#### 4.1.4. Preventing and Treating Sores from Urine and or Fecal Leakage

Water scarcity has compounded the misery of some of our study participants, as observed by previous researchers [17,39]. Ideally, community-based programs such as pump maintenance and improving water access [40] would be rolled out to reach out to every rural Malawian setting. This would not only help women living with incontinence manage and cope with their condition, but also promote hygiene and prevent diseases such as diarrhea. Our study participants were less likely to look for medical treatment for their sores, as they opted to use home remedies, aligning with findings from other studies [28,32,41]. According to the Geneva Foundation for Medical Education and Research [42], genital sores, abrasions, and skin rash from fistula incontinence can best be managed by sitzs baths and zinc oxide barrier ointment, yet none of our participants mentioned using such measures for their skin complications.

An explanation could be that, since treatment for their fistula condition was unavailable, it might have discouraged them from seeking treatment for their skin complications. Fear of being stigmatized or labeled as having a sexually transmitted infection (STI) could have also deterred them from seeking medical care [43]. It is an opportunity for community healthcare workers to sensitize the community on causes and debunk myths and misconceptions, while at the same time normalizing the fistula as any other condition. Additionally, they could provide the affected women with zinc oxide barrier ointments and counseling on sitzs baths to minimize unnecessary suffering.

### 4.2. Emotional-Focused Coping

The emotional-focused coping strategies used by participants in our study are similar to those reported by earlier studies [28,30,39].

#### 4.2.1. Spouse and Family Support

The importance of family and spouse support in coping with such a devastating condition as fistula cannot be overemphasized. Family, spousal, and social support could be a source of strength and self-confidence [19]. In Malawi, Yeakey et al. (2009) reported high spouse and family support which encouraged health care seeking and reintegration of the affected women into their communities [20]. Similarly, in Tanzania, Pope et al. (2011) reported an easy transition home and acceptance in their communities after surgery for women with family and social support, as opposed to no support [44]. Interventions to prevent, treat, and reintegrate women with fistula, should take into account and accentuate the importance of family, spousal, and social support in such interventions.

#### 4.2.2. Having Faith in God as a Source of Encouragement

The significance of religion in dealing with stigmatized conditions such as fistula need not be underestimated. In the Bible, lepers were isolated, excommunicated and viewed as unclean, and untouchable, and were isolated from the rest of the people, until they were healed. It was the religious leaders who examined and confirmed whether the lepers were cured and fit for re-integration [45]. Women with fistula are to some extent ‘modern day lepers’ in the sense that they may not necessarily be excluded by their community as observed in this study, but they deem themselves as unclean and therefore exclude themselves. Considering that women in this study were using their faith in God to cope with their condition, if religious leaders are educated accordingly, they could be used to further help these women get guidance and help on how to deal with their condition [46].

#### 4.2.3. Children Rearing as a Source of Motivation

In Malawi, as with other African societies, a woman’s status (or worth) could be determined by her ability to bear children and raise a family [19,30,38]. Therefore, having children increases the woman’s value and status in society, and may determine the outcome of marriages [19,47]. This may explain why some of our study participants felt encouraged by the fact that they had children and/or continued to bear children after fistula. Furthermore, it may also explain why most women remained married.

### 4.3. Study Strength and Limitations

Our study participants were women waiting for repair of fistula or recovering from surgery, which gave us an opportunity to gain insights from their lived experiences of coping with incontinence. Our study was both institutional and community based, which helped to give a voice to a wide range of affected women, including those women who could not afford transport to seek care at the Fistula Care Center. Findings of our study add to existing knowledge on the coping mechanisms of women living with consequences of obstetric fistula. We provided rich detailed descriptions of the research methodology to determine transferability to other similar settings.

However, our study was not without limitations. Employing the snow ball sampling technique could have caused selection bias, limiting the selection to only the ones the women themselves knew about and not those who might have effectively hid themselves or “pass” for normal, potentially leaving out some of the most affected women who might have very different stories and perspectives to share. Twenty of the participants in our study were interviewed at the Fistula Care Center, which could have affected their responses. To avoid social desirability, the interviewer was not one of the health workers at the Fistula Care Center, and voluntary participation was emphasized. Our study population was limited to the views of patients only, excluding health care providers. Hence, we could not determine whether there were standard operating procedures or policies on the type of information given to women about obstetric fistula as regards self-management. Future research may consider exploring this area, to glean a more comprehensive picture of obstetric fistula’s clinical and self-management.

## 5. Conclusions

As women living with incontinence from obstetric fistula await repair, they employ different methods of coping, some of which conflict with proper bladder management, indicating a lack of proper counseling on self-management of incontinence and its consequences. Providing women with adequate information on self-management and the availability of treatment and support for fistula sequelae could help to ensure physical and emotional comfort. All have an active role to play, be they within the women’s homes, communities, religious institutions and/or facilities.

## Figures and Tables

**Figure 1 ijerph-16-03092-f001:**
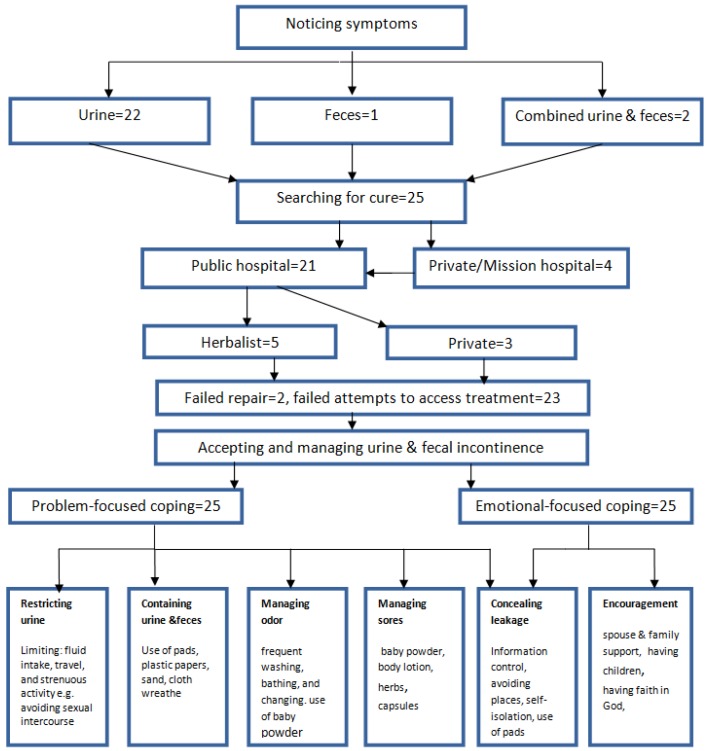
Obstetric fistula pathway delineated for the 25 study participants based on their testimonies.

**Figure 2 ijerph-16-03092-f002:**
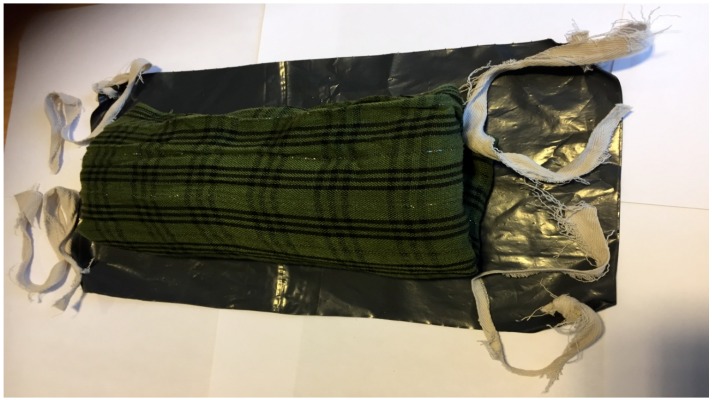
Showing a folded cloth pad on a water proof black plastic sheet. Picture taken with patient’s consent.

**Figure 3 ijerph-16-03092-f003:**
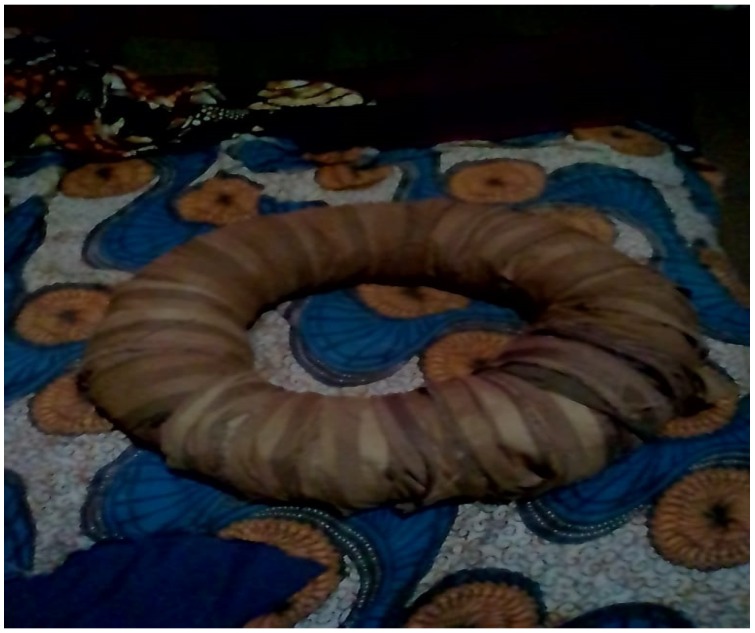
An example of a cloth wreathe that was used to limit the flow of urine. Picture taken with patient’s consent.

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
