# Peer review of "Coping with a Disruptive Life Caused by Obstetric Fistula: Perspectives from Malawian Women"

_ijerph, 2019, doi:10.3390/ijerph16173092_

Round 1

Reviewer 1 Report

Thank you for giving me the opportunity to review this interesting study. The topic is important, and gives a new knowledge, particular an interesting account of coping strategies.

However, I think a few aspects could be improved.

In the abstract, and also the introduction there are a few phrases where information is duplicated.

I would have expected in the introduction to find the only systematic review from Adler et al. 2013. I’m not sure if the DHS really reports fistula prevalence or incontinence (which can also have other reason). To my knowledge the DHS does not include a physical examination? But can be wrong.

Also, be aware that there ae increasingly reports that indicate that fistulas are the result of complications while doing a Caesarean section (e.g. undetected bladder injuries).

The introduction could be a bit better structured, e.g. para 3 and para 6 both touch on the lack of surgery.

The method section misses a few information: why did you include women admitted for fistula repair and from the community. I’m not saying this is wrong, but an explanation is missing.

I expect a bit more information on who (qualification, etc) did the interview, how (place, comfort, time etc), how the sample size was determined. Although you cite another paper, I think the important details also need to be here.

The figure which provides an overview of the results is very nice, but the layout can be improved as arrows are a bit crossing unnecessarily long distances. I assume that the name of the participants are not the correct names (which they should not for confidentiality)

I assume the results section can be shortened a bit, some quotes are overlapping

The discussion could work out a bit more: what is new. I’m also a bit surprised that there is no discussion on the lack of access to repair. So many women had their fistula for such a long time, and went to seek care, but obviously no one could direct them?  Apart from this, I feel the discussion could be a bit condensed. I’m not sure I understand the reference to lepers. Also, it looks like that the women are in fact not excluded. In response I also do not understand the role of religious leaders?

Author Response

26th July, 2019

Manuscript ID: ijerph-513889
Type of manuscript: Article
Title: Coping with a disruptive life caused by obstetric fistula: Perspectives from Malawian women

Response to the reviewers' comments

Dear Editor,

We are grateful to the reviewers for their time and constructive comments on our revised manuscript referenced above. We provide a point by point response explaining how we have addressed each of the reviewers′ comments as follows:

REVIEWER 1

Comments and Suggestions for Authors

Thank you for giving me the opportunity to review this interesting study. The topic is important, and gives a new knowledge, particular an interesting account of coping strategies.

However, I think a few aspects could be improved.

1. In the abstract, and also the introduction there are a few phrases where information is duplicated.

Response: Thank you for your observation. Please take note that this issue has been addressed. Please check the abstract and the introduction section.

2. I would have expected in the introduction to find the only systematic review from Adler et al. 2013. I’m not sure if the DHS really reports fistula prevalence or incontinence (which can also have other reason). To my knowledge the DHS does not include a physical examination? But can be wrong.

Response: We agree with your comment. Therefore, a sentence has been added to address this concern. As well as the systematic review in question. Please see introduction, page 1, line 35-41 and page 2, line 1-6.

3. Also, be aware that there are increasingly reports that indicate that fistulas are the result of complications while doing a Caesarean section (e.g. undetected bladder injuries).

Response: This observation is also right, thank you. We have addressed this concern accordingly.

4. The introduction could be a bit better structured, e.g. para 3 and para 6 both touch on the lack of surgery.

Response: The paragraphs in question have been revised accordingly. The phrase in question has been deleted from paragraph 6. See introduction page 2, line 29-34.

5. The method section misses a few information: why did you include women admitted for fistula repair and from the community. I’m not saying this is wrong, but an explanation is missing.

Response: Thank you for your observation. We have included extra information in the methods section to address this issue. See methods page 2, line 39-43.  

6. I expect a bit more information on who (qualification, etc) did the interview, how (place, comfort, time etc), how the sample size was determined. Although you cite another paper, I think the important details also need to be here.

Response: The information in question has been added accordingly. See methods section page 2, line 43 and 44; line 47 and 48; page 3, line 1-4.

7. The figure which provides an overview of the results is very nice, but the layout can be improved as arrows are a bit crossing unnecessarily long distances.

Response: Thank you for your suggestion. We have restructured the figure accordingly. Please see figure 1, page 4.

8. I assume that the name of the participants are not the correct names (which they should not for confidentiality).

Response: We used pseudonyms for participants in our manuscript, we just forgot to mention before, a statement has now been added in the methods section. See methods, page 3, line 23 and 24.

9. I assume the results section can be shortened a bit, some quotes are overlapping

Response: Some quotes have been removed accordingly. Thank you.

10. The discussion could work out a bit more: what is new. I’m also a bit surprised that there is no discussion on the lack of access to repair. So many women had their fistula for such a long time, and went to seek care, but obviously no one could direct them?  Apart from this, I feel the discussion could be a bit condensed. I’m not sure I understand the reference to lepers. Also, it looks like that the women are in fact not excluded. In response I also do not understand the role of religious leaders?

Response: Thank you for your observation. Firstly, the discussion on lack of access to repair was deliberately excluded in this paper, because the issue was already discussed in our previously published paper. This was done to avoid an overlap between the two papers. Please advise if it is okay to include it in this paper. 

Secondly, we have revised the section concerning the reference to lepers to help clarify what we meant.

Please do not hesitate to let us know if there is anything else that needs to be improved.

Thank you for your support.

Sincerely,

Josephine Changole (Mrs.)

Reviewer 2 Report

The manuscript by Changole et al. targets methods of dealing with problems connected with urinary and fecal incontinence in Malawian women with obstetric fistula.

This paper is based on the same population of women, which was presented in the authors' previous study (ref. No 11). The previous paper included the detailed characteristics of 25 women with obstetric fistula. Unfortunately, both manuscripts are very similar. The current manuscript manages the topic of coping with urinary incontinence in slightly more detailed.

The manuscript is based on statements of selected women with obstetric fistula. The presented problems of women with obstetric fistula result from low level of medical care, sanitary difficulties and limited number of qulified surgeons.

Descriptions of Figure 1 are incomplete (line 24).

The authors included photos of makeshift sanitary materials.

I did not find any statement of a woman with fecal incontinence.

The manuscript's references are not up-to-date. The two most recent references are the athors' previous publications (ref. No 11 from 2017 and ref. No 42 from 2018).

Author Response

26th July, 2019

Manuscript ID: ijerph-513889
Type of manuscript: Article
Title: Coping with a disruptive life caused by obstetric fistula: Perspectives from Malawian women

Response to the reviewers' comments

Dear Editor,

We are grateful to the reviewers for their time and constructive comments on our revised manuscript referenced above. We provide a point by point response explaining how we have addressed each of the reviewers′ comment as follows:

REVIEWER 2

Comments and Suggestions for Authors

The manuscript by Changole et al. targets methods of dealing with problems connected with urinary and fecal incontinence in Malawian women with obstetric fistula.

This paper is based on the same population of women, which was presented in the authors' previous study (ref. No 11). The previous paper included the detailed characteristics of 25 women with obstetric fistula. Unfortunately, both manuscripts are very similar. The current manuscript manages the topic of coping with urinary incontinence in slightly more detailed.

Response: We agree with your observation and we are aware of the similarities. As we stipulated in the methods section that the paper is based on a subset of data on a major study on experiences of Malawian women living with obstetric fistula. As such, the sample, setting, and the methods are the same. However, as you have rightly pointed out, the focus of this paper was the coping strategies that the women used. Thank you. 

The manuscript is based on statements of selected women with obstetric fistula. The presented problems of women with obstetric fistula result from low level of medical care, sanitary difficulties and limited number of qualified surgeons.

1. Descriptions of Figure 1 are incomplete (line 24).

Response: The description of the figure in question has been revised accordingly. See figure 1, page 4, line 4.

The authors included photos of makeshift sanitary materials.

2. I did not find any statement of a woman with fecal incontinence.

Response: In the figure one we indicated that there was one participant with fecal incontinence, and two participants with combined urine and feces, and they used  coping strategies similar to those with urinary incontinence. However, we have included a quote to address this concern. See findings, page 5, line 12-17.  Thank you.

3. The manuscript's references are not up-to-date. The two most recent references are the authors' previous publications (ref. No 11 from 2017 and ref. No 42 from 2018).

Response: Thank you for your observation. We have added new references to address this concern. Please check the reference list page 12-15.

Please do not hesitate to let us know if there is anything else that needs to be improved.

Thank you for your support.

Sincerely,

Josephine Changole (Mrs.)

Round 2

Reviewer 2 Report

Dear Authors - Thank you for attempting to address most of my concerns.

I accept the manuscript in the present form.